# Hypnozoite depletion in successive *Plasmodium vivax* relapses

**Rintis Noviyanti[1], Kelly Carey-Ewend**  **[2,3]\*, Leily Trianty[1], Christian Parobek[3], Agatha Mia Puspitasari[1], Sujata Balasubramanian[3], Zackary Park[3], Nicholas Hathaway[4], Retno A. S. Utami[1], Saraswati Soebianto[5], Jeny Jeny[5], Frilasita Yudhaputri[1], Aditya Perkasa[1], Farah N. Coutrier[1], Yusrifar K. Tirta[1], Lenny Ekawati[5], Bagus Tjahyono[6], Inge Sutanto[7], Erni J. Nelwan[7], Herawati Sudoyo[1], J. Kevin Baird[1,7], Jessica T. Lin**  **[3]\***

**1** Eijkman Institute for Molecular Biology, Jakarta, Indonesia, **2** Gillings School of Global Public Health, University of North Carolina, Chapel Hill, North Carolina, United States of America, **3** University of North Carolina School of Medicine, Chapel Hill, North Carolina, United States of America, **4** University of Massachusetts Chan Medical School of Medicine, Worcester, Massachusetts, United States of America, **5** Eijkman-Oxford Clinical Research Unit, Jakarta, Indonesia, **6** Health Services, Army of the Republic of Indonesia, Jakarta, Indonesia, **7** Faculty of Medicine, University of Indonesia, Jakarta, Indonesia

\* kelly_carey-ewend@med.unc.edu (KCE); jessica_lin@med.unc.edu (JTL)

**Data Availability Statement:** All pvsmp1 haplotype sequences can be found via GenBank accession numbers OM310778-OM310805. Data on

## Abstract

Genotyping *Plasmodium vivax* relapses can provide insights into hypnozoite biology. We performed targeted amplicon sequencing of 127 relapses occurring in Indonesian soldiers returning to malaria-free Java after yearlong deployment in malarious Eastern Indonesia. Hepatic carriage of multiple hypnozoite clones was evident in three-quarters of soldiers with two successive relapses, yet the majority of relapse episodes only displayed one clonal population. The number of clones detected in relapse episodes decreased over time and through successive relapses, especially in individuals who received hypnozoiticidal therapy. Interrogating the multiplicity of infection in this *P. vivax* relapse cohort reveals evidence of independent activation and slow depletion of hypnozoites over many months by multiple possible mechanisms, including parasite senescence and host immunity.

## Author summary

Investigating relapse patterns in infections of *Plasmodium vivax*, a parasite that causes malaria, is challenging due to concurrent reinfection events alongside true relapses in most clinical cohorts. We performed sequencing on *P. vivax* samples from a cohort of Indonesian soldiers who were exposed to the parasite while deployed in a malaria-endemic region and then experienced relapses after their return to a region with no current malaria transmission. From these true relapses, we show that most infected individuals harbor multiple lineages of hypnozoites (latent liver stage parasites that reactivate to cause relapse) but individual relapses are largely driven by a single hypnozoite lineage or clone. Additionally, the average number of parasite clones detected in each relapse decreases over time. These findings suggest that *P. vivax* hypnozoites activate

haplotypes and relapses are available in the manuscript and its supporting information files.

**Funding:** This work was supported by the Wellcome Trust Africa Asia Program Vietnam (https://wellcome.org/what-we-do/our-work/programmes-and-initiatives-africa-and-asia), Medicines for Malaria Venture (https://www.mmv.org/), Ministry of Research and Technology - the Republic of Indonesia, the Alumni Grant Scheme - the Australia Awards in Indonesia (https://australiaawardsindonesia.org/) to RN, and the National Institute of Allergy and Infectious Diseases (https://www.niaid.nih.gov/) at the National Institutes of Health through grants K08AI110651 and R21AI152260 to JTL. The funders had no role in the study design, data collection or analysis, decision to publish, or preparation of the manuscript.

**Competing interests:** The authors have declared that no competing interests exist.

independently from each other and that their population in the liver decreases over time after the initial infection, possibly due to immune clearance or loss of parasite viability.

## Introduction

Over one-third of the world population remains at risk for *Plasmodium vivax*, the most widely distributed of the human malarias [1]. The World Health Organization estimated 4.6 million cases of vivax malaria worldwide in 2020 [2], but this likely belies a much larger asymptomatic reservoir of persons harboring sub-patent infections as well as latent liver-stage hypnozoites that are not cleared by therapy targeting the blood stages. Hypnozoite carriers may relapse weeks to several years after primary infection. This latent reservoir imposes a difficult challenge to malaria control efforts.

Hypnozoite-induced relapses are responsible for most of the burden of acute *P. vivax* in Southeast Asia and South Pacific [3,4], but the molecular characterization of relapse events remains limited. This is because parasitemias due to relapse and reinfection occur contemporaneously and cannot be differentiated. Studies of relapse free of that ambiguity require relocation of infected persons to settings where there is no or very low risk of mosquito-borne transmission. Genotyping in this scenario has revealed that relapses are frequently but not always polyclonal (contain multiple malaria strains), and may contain strains that are either identical to those detected in the preceding infection episode (homologous), or different/novel (heterologous) [5–8].

In this study, we apply amplicon deep sequencing to study relapses among Indonesian soldiers returning to a malaria-free area of Java after year-long deployment in malarious Indonesian Papua. The cohorts studied are unique because new parasitemias among them could be classified as relapses with a high degree of confidence. We found that the multiplicity of infection—the number of strains or parasite variants detected in the blood in a single infection episode- decreased over time, suggesting that this may be a proxy for hypnozoite load in the liver, with slow depletion over many months.

## Methods

### Ethics statement

The study was approved by the Ethics Committee of the Faculty of Medicine, University of Indonesia (ref.no. 13/H2.F1/ethic/2013, 9,10] and the Institutional Review Board of the University of North Carolina, Chapel Hill (Study #16–0079). Written informed consent was obtained from all participants at enrollment.

### Study population

The study cohort was derived from two drug trials investigating the anti-relapse efficacy of primaquine paired with different schizonticidal treatment regimens [9,10]. Study participants were male Indonesian soldiers, aged 21–50 years old, returning to Java (either Lumajang or Sragen) after being stationed in Papua, Indonesia for roughly one year. Subjects did not receive malaria prophylaxis during their deployment as per Indonesian medical army doctrine but received routine health screening upon arriving in Java and were treated with dihydroartemisinin-piperaquine if found to be malaria-positive by microscopy. Of note, Papua, where their exposures occurred, has endemic *P. vivax* transmission with an attack rate among deployed Indonesian soldiers previously estimated at 2 vivax cases/person-year [11], while Java no

longer has endemic malaria transmission, excepting a few well known persisting foci [12]. Unfortunately, the number and timing of acute vivax malaria attacks prior to disembarkment is not known; such attacks would have been treated with dihydroartemisinin-piperaquine. Screening began one month after arrival, with enrollment of those positive for vivax malaria by microscopy. Given the lack of local malaria transmission and the infrequent carriage of asymptomatic parasitemia in this population, malaria cases detected at enrollment are largely, if not all, attributable to hypnozoite-borne relapse rather than recent mosquito-borne reinfection or chronic parasitemia. Therefore, *P. vivax* cases found at initial screening following study enrollment are hereafter referred to as "first relapse". These subjects were randomized to malaria treatment with or without primaquine for 14 days (30mg base) and subsequently followed for 12 months to survey for another relapse, either via microscopic screening at regular intervals (weekly for 10 weeks then 4 more times thereafter) or via passive symptomatic case detection with confirmatory microscopy. Any subsequent *P. vivax* cases identified thus are hereafter referred to as the "second relapse". The results from a study in Cambodia that examined *P. vivax* relapses using the same genetic methodologies described below, are also referenced in the results section, though this study relied on genotyping of parasites in order to differentiate relapses from reinfections rather than presence of the cohort in a location with no endemic malaria transmission [12].

## Amplicon deep sequencing and pvmsp1 haplotype determination

Parasite DNA extracted from blood samples collected at the time of relapse was used for targeted amplicon deep sequencing at a 117–base pair variable portion of the 42-kDa region of *pvmsp1* as previously described [13]. Briefly, nested PCR incorporating a multiplex identifier barcoded sequence was performed for each sample in duplicate, then amplified products were purified, pooled, and sequenced on the Ion Torrent platform. Analysis of sequences was done using SeekDeep version 2.5.0 (http://seekdeep.brown.edu/) to identify haplotypes that differed by at least 1 single nucleotide polymorphism (SNP) from one another, and appeared at >1% frequency within isolates [14]. Additionally, haplotypes with only 1 SNP difference within a single sample were collapsed, by allowing 1 high-quality base mismatch during clustering between the 2 replicates for each sample, so as to err in favor of stringency in calling haplotypes.

## Data analysis

Multiplicity of infection (MOI) was defined as the number of unique *pvmsp1* haplotypes detected in a single relapse sample. DNA alignment of haplotypes was generated using GenVison software (DNAStar, Madison, WI). *Pvmsp1* genetic diversity in the sample cohort was estimated by calculating the expected heterozygosity ($H_E$) [15]. Differences in MOI between first and second relapses as well as early and late relapses were tested using Pearson's chi-square analysis and Kruskal-Wallis rank sum testing in the presence of non-normality. Linear regression was used to evaluate MOI as a function of days from enrollment to relapse, and change in MOI from first to second relapse. Statistical analyses were performed using Graph-Pad Prism 9.0.

## Results

### Genetic diversity of *P. vivax* relapses in return soldiers

Single amplicon deep sequencing of relapsing *P. vivax* revealed many monoclonal infections despite overall high genetic diversity. In total, parasite DNA from 127 relapse samples from 94

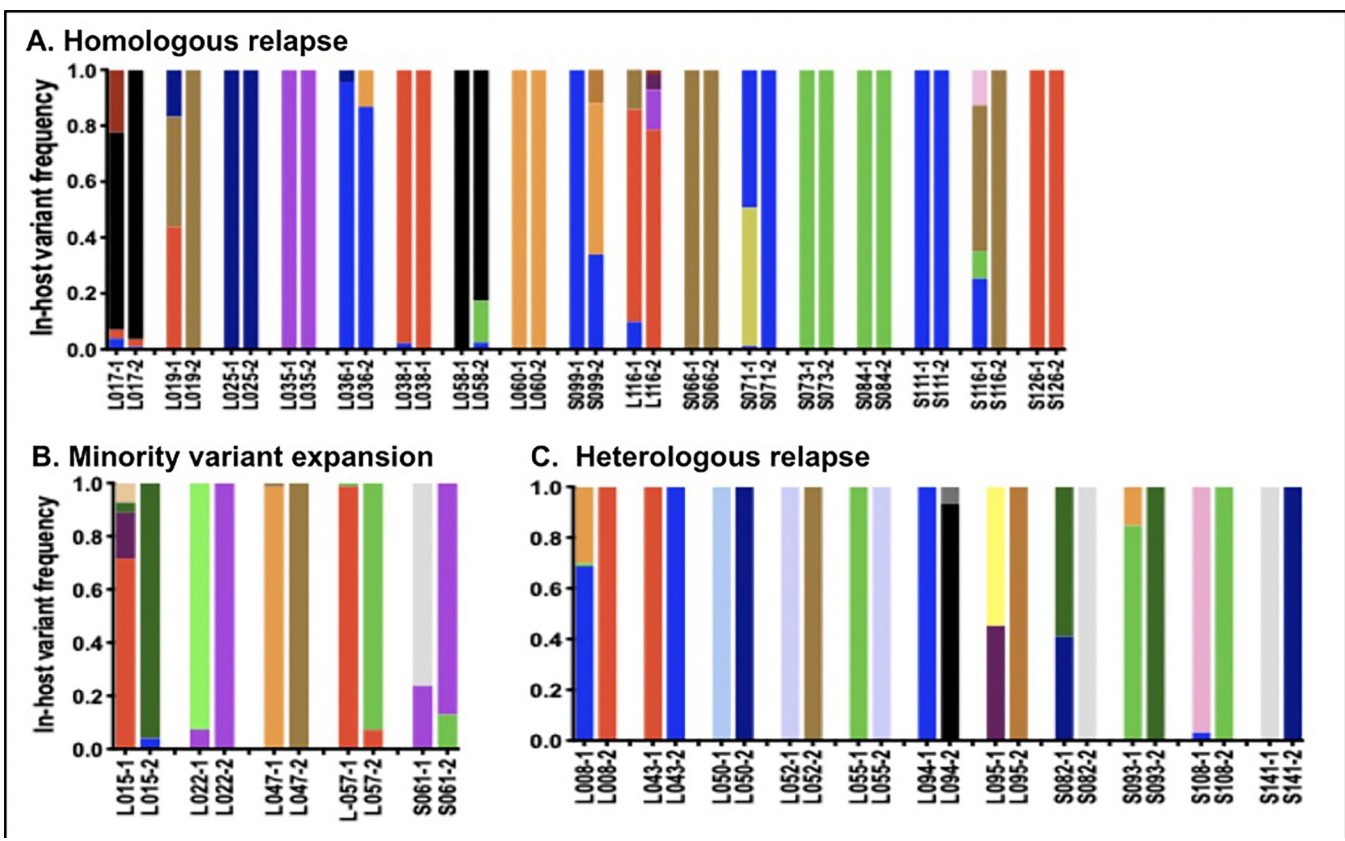

**Fig 1. Genotypes of 33 pairs of relapses in returning Indonesian soldiers based on *pvmsp1* amplicon deep sequencing.** The in-host frequency of different *pvmsp1* haplotype variants is displayed for the first (-1) and second relapse (-2) in each individual. 17 pairs showed a homologous relapse pattern (A), 5 demonstrated minor variant expansion (B), and 11 were heterologous relapses (C). Though 61% of relapses were monoclonal, only 24% had a single variant detected across both relapses. Examples of each relapse pattern are shown in (D).

soldiers (74 returning to Lumajang and 20 returning to Sragen) were successfully sequenced at *pvmsp1* with a median read depth of 4,008 (IQR 2442–5755) (S1 Fig). Sequencing revealed 44 variable sites within the 117bp amplified region, the majority of which were nonsynonymous substitutions, and altogether yielded 28 unique haplotypes (*pvmsp1* variants) across the 127 isolates (GenBank accession numbers OM310778-OM310805) (S2 Fig). Three common haplotypes appeared in at least 15% of individuals, while 9 only occurred in a single sample. 18/28 haplotypes blasted to previously published sequences. Expected heterozygosity ($H_E$) was high using a single marker ($H_E$ = 0.77 among 94 first-relapse samples, reflecting an average 77% probability that 2 samples taken at random from the population will display different *pvmsp1* haplotypes).

Despite this relatively high diversity, the majority of relapses were monoclonal, with 61% (77/127) of relapses showing only one *pvmsp1* haplotype (**Fig 1**). Up to 4 *P. vivax* clones were detected in the rest of the relapse samples, with a mean multiplicity of infection of 1.6 overall.

## Clonal patterns of relapsing *P. vivax*

While individual relapse episodes were monoclonal in nature, individuals often harbored multiple parasite variants across their relapses. Among the 94 individuals, 33 (17 of whom received primaquine) had two successfully sequenced relapse episodes during study follow-up. Of the 66 relapse episodes among these 33 individuals with successive relapses, 61% (40/66) were

monoclonal with only one haplotype detected. However, when examining both relapse episodes of the 33 individuals, just 24% (8/33) of individuals had only one haplotype detected across both of their relapses, while 76% (21/33) had multiple haplotypes present across both relapses.

Additionally, the clonality of relapses generally decreased over time. Among all first relapses, 55% (47 out of 85) were monoclonal, while 71% (30/42) of the second relapses were monoclonal (p = 0.05). Among all 127 relapses, the average multiplicity of infection (MOI) of relapses within the first 30 days after enrollment ("early relapses") was 1.8 while the average MOI of relapses occurring more than 60 days after enrollment ("late relapses") was 1.3 (p = 0.02). Across this entire cohort that was not exposed to new infectious mosquito bites after their return to Java, the clonality of relapses decreased on a time scale of approximately 1 clone every 333 days, based on simple linear regression (**Fig 2A**).

When examining change in MOI from first to second relapse in each individual, decreasing clonality over time is also apparent. Among the 33 returning Indonesian soldiers with dual relapse, 13 displayed fewer variants at the second relapse, 16 had the same number of variants (13 of which displayed only monoclonal infections), and only 4 displayed an increased number of variants at their second relapse. This decrease in clonality from first to second relapse was driven by those treated with primaquine at the first relapse (blue trendline in **Fig 2B**). However, a parallel analysis of previous *pvmsp1* amplicon sequencing data from paired relapses among 18 Cambodian soldiers not treated with primaquine [13], shows a similar trend (black trendline in **Fig 2B**). This Cambodian soldier cohort showed a higher *P. vivax* diversity overall: 34 *pvmsp1* haplotypes were identified among 107 samples, with a mean MOI of 2.9 (range 1–8) using updated Seekdeep clustering parameters similar to that used for the present Indonesian cohort.

Regarding relapse patterns, just over half of consecutive relapse episodes were homologous in nature. Comparing the sets of haplotypes present in each individual's pair of relapses, 52% had a homologous relapse pattern (dominant haplotype was the same in both relapses), 15% displayed minority variant expansion whereby a variant with <20% in-host frequency in the first relapse reappeared as the dominant haplotype in the second relapse, and 33% displayed a heterologous relapse pattern without any shared variants (**Fig 1**). The median time between detection of the first and second relapses was 55 days (range 17 to 270). There was no

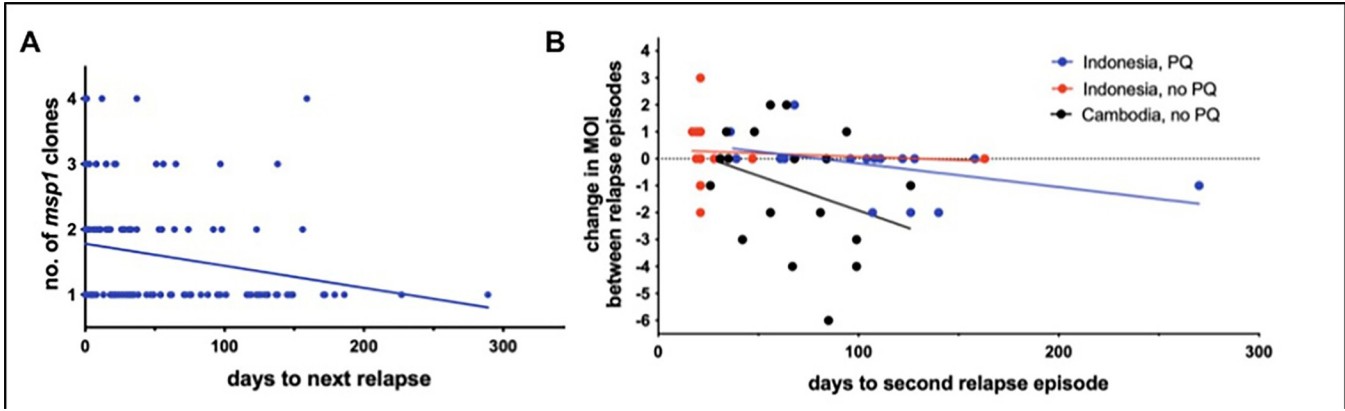

**Fig 2. Decreased clonal burden in relapses over time.** (A) Among 127 relapses in returning Indonesian soldiers, the number of *pvmsp1* clones detected at relapse decreased over time, as measured by days since enrollment one month after arriving to Java (slope = -0.003, p = 0.01). (B) Among 33 Indonesian soldiers with dual relapse, the multiplicity of infection (MOI) decreased between the first and second relapse episode, driven by those who received primaquine at their first relapse (depicted in blue) (slope = -.007, p = 0.03). Similarly, relapses in 18 Cambodian soldiers determined genotypically to have relapse [13] suggest a decrease in MOI from their initial vivax episode to relapse (slope = -0.03, p = 0.20). Lines of best fit were determined by linear regression.

association between the number of days between relapses and the pattern of relapse (median 76 days vs 61 days for homologous vs. heterologous relapse pairs, p = 0.79).

## Discussion

Our use of amplicon deep sequencing to investigate the genetic diversity and multiplicity of infection of *P. vivax* relapses in a cohort of Indonesian soldiers returning to a non-endemic area yielded interesting observations pertinent to hypnozoite biology.

First, although the majority of subjects (76%) harbored multiple parasite strains in their livers (detected through the presence of multiple haplotypes across their relapses), 61% of individual relapses were monoclonal. This suggests individual hypnozoite clones (and probably individual hypnozoites) activate independently of other clones present in the same individual. This is not surprising considering the relatively few hypnozoites resident in the liver among billions of hepatocytes [16]. However, it stands in contrast to the predominately polyclonal relapses described previously in individuals living in but then removed from endemic areas [5,6,13]. We believe this likely reflects lower hypnozoite loads in our cohort, akin to the Australian soldiers previously described to have suffered monoclonal relapses (85/86 relapse isolates) after deployment in East Timor, in contrast to the East Timorese resident population who were more likely to harbor polyclonal infection [8]. When multiple relapses were observed in the Australian soldier cohort, they too more frequently showed heterologous genotypes. The major difference was the receipt of prophylaxis both during deployment and terminal prophylaxis at the end of deployment, which likely led to the much longer time interval to relapse (median 181 days vs. 30 days in the Indonesian soldiers who did not receive chemoprophylaxis). The less frequent polyclonal relapses in these returning soldier cohorts could also be due to the absence of hypothesized environmental triggers for hypnozoite activation related to malaria exposure (e.g. infective vector bites or malaria superinfection), reducing the chance that multiple clones will reactivate simultaneously.

Second, we found that the multiplicity of infection (MOI) of *P. vivax* relapses decreased over time, both within the cohort as a whole and within individuals with consecutive relapses. This suggests that MOI may serve as a proxy indicator for hypnozoite burden in the liver, and hypnozoite depletion through reactivation, hepatocyte turnover, senescence, and/or immunity, manifests as a decline in MOI over time. A faster decline in those treated with primaquine (**Fig 2B**) supports this interpretation, and an overall slow rate of decline when few hypnozoites remain, also suggested by our data, has previously been modeled [16]. Strain-specific acquired immunity and subsequent clonal suppression have been intuited in case studies of relapse [6,17]. This mechanism could also account for the pattern of minority variant expansion seen from one relapse to the next that we and others have described [5,13].

Our findings may be limited by a partial capture of all relapses. Subpatent relapses may have been missed because enrollment and subsequent case detection relied on microscopy, with PCR used only for confirmation of positive cases, and the frequency of follow-up decreased after the first 10 weeks of surveillance. Thus, it is possible that hypnozoite activation occurred more frequently, with only a subset of relapses reaching microscopic patency. This possibility of unknown likelihood does not diminish the observed decline in MOI with duration of interval between parasitemia events in our subjects. Further, as the cohort derived from primaquine treatment trials, 17 out of 33 individuals with dual relapses received hypnozoiticidal therapy and the reduction in MOI in subsequent relapses among these individuals cannot be separated from the drug effect of the therapy they received. However, the presence of this same trend in the prior Cambodian cohort that did not receive primaquine suggests that the pattern is not solely attributable to hypnozoiticidal therapy. Lastly, single amplicon deep

sequencing is sensitive to detecting minority variants but limited in its ability to infer genetic relatedness between different haplotypes. In particular, relapses that appear heterologous could in fact be related as meiotic siblings derived from the same ookinete in the same mosquito. It is possible that the proportion of relapses deemed monoclonal might decrease with the use of additional markers, but, given the genetic diversity revealed at *pvmsp1*, we believe this difference would be incremental and not change our overall findings.

Future studies that track participants from the time of primary infection and ideally capture more relapses are needed to confirm our findings. If multiplicity of infection is indeed a proxy indicator for latent hypnozoite burden, this metric could be useful for understanding hypnozoite biology, modeling the impact of interventions on the vivax reservoir, and perhaps evaluating dosing strategies for achieving radical cure in different populations [18].

Preliminary findings were previously presented at the 2018 Annual Meeting of the American Society of Tropical Medicine and Hygiene as part of a symposium entitled, "Unraveling the biology of the hypnozoite—integrating findings from lab models and field studies of Plasmodium vivax."

## Supporting information

**S1 Fig. Sample analysis flow diagram.** All samples available from the Lumajang cohort were selected for sequencing, while only samples from individuals with dual relapse were selected from the Sragen cohort. The final cohort used for sample analysis represented 62% (127/206) of the original sample set, with equal attrition from both cohorts.
(TIF)

**S2 Fig.** (A) DNA alignment of the 28 *pvmsp1* haplotypes detected in the 127 samples from 94 individuals. Each concentric ring represents a unique sequence, differing by at least one single-nucleotide polymorphism at one of 44 variable sites detected within the 117 bp amplicon. Nucleotides are represented by different colors (adenine, red; thymine, blue; cytosine, green; and guanine, yellow). (B) Frequency of unique *pvmsp1* haplotypes within the study population (out of 127 isolates). The red portions of the columns represent the proportion that occurred as a minority variant (existing at 1–20% frequency within the individual isolate). The three most common haplotypes were detected in at least 15% of all samples and 9 haplotypes were singletons, found in only one sample each. (C) Alignment of all 28 haplotypes of *pvsmp1*.
(TIF)

**S1 Data. Microsoft Excel spreadsheet with three tabs containing data and data analysis.** The first shows the counts and distribution of all 28 detected *pvsmp1* haplotypes, the second shows the haplotypes by relapse, and the third gives information about each detected relapse.
(XLSX)

## Acknowledgments

We thank Dorothya R. Lestari, the study participants, and the study staff at the Indonesian Armed Forces and Eijkman-Oxford Clinical Research Unit for their support.

## Author Contributions

**Conceptualization:** Rintis Noviyanti, J. Kevin Baird, Jessica T. Lin.

**Data curation:** Christian Parobek.

**Formal analysis:** Kelly Carey-Ewend, Christian Parobek, Zackary Park, Jessica T. Lin.

**Funding acquisition:** Jessica T. Lin.

**Investigation:** Rintis Noviyanti, Leily Trianty, Christian Parobek, Agatha Mia Puspitasari, Sujata Balasubramanian, Retno A. S. Utami, Saraswati Soebianto, Jeny Jeny, Frilasita Yudhaputri, Aditya Perkasa, Yusrifar K. Tirta, Lenny Ekawati, Bagus Tjahyono, Jessica T. Lin.

**Methodology:** Sujata Balasubramanian, Nicholas Hathaway.

**Project administration:** Rintis Noviyanti, Farah N. Coutrier, Inge Sutanto, Erni J. Nelwan, Herawati Sudoyo, Jessica T. Lin.

**Resources:** Rintis Noviyanti, J. Kevin Baird, Jessica T. Lin.

**Software:** Christian Parobek, Zackary Park.

**Supervision:** Rintis Noviyanti, Christian Parobek, Farah N. Coutrier, J. Kevin Baird, Jessica T. Lin.

**Visualization:** Jessica T. Lin.

**Writing – original draft:** Kelly Carey-Ewend, Jessica T. Lin.

**Writing – review & editing:** Rintis Noviyanti, Christian Parobek, J. Kevin Baird, Jessica T. Lin.

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
