## [Decision Letter · Decision Letter 0]

17 Jun 2022

Dear Mr. Carey-Ewend,

Thank you very much for submitting your manuscript "Hypnozoite depletion in successive Plasmodium vivax relapses" for consideration at PLOS Neglected Tropical Diseases. As with all papers reviewed by the journal, your manuscript was reviewed by members of the editorial board and by several independent reviewers. The reviewers appreciated the attention to an important topic. Based on the reviews, we are likely to accept this manuscript for publication, providing that you modify the manuscript according to the review recommendations. 

The manuscript has been assessed by 3 reviewers and while there is lack of novelty in the findings, the reviewers generally agree that these kinds of studies (clinical gametocyte field studies) are challenging to conduct and validation of earlier studies does lend importance to the current study. Hence, there is general consensus that this work should be accepted after the authors address the queries raised by the reviewers.

Sincerely,

Kevin Shyong-Wei Tan

Associate Editor

Mary Lopez-Perez

Deputy Editor

The manuscript has been assessed by 3 reviewers and while there is lack of novelty in the findings, the reviewers generally agree that these kinds of studies (clinical gametocyte field studies) are challenging to conduct and validation of earlier studies does lend importance to the current study. Hence, there is general consensus that this work should be accepted after the authors address the queries raised by the reviewers.

Reviewer's Responses to Questions

**Key Review Criteria Required for Acceptance?**

**Methods**

-Are the objectives of the study clearly articulated with a clear testable hypothesis stated?

-Is the study design appropriate to address the stated objectives?

-Is the population clearly described and appropriate for the hypothesis being tested?

-Is the sample size sufficient to ensure adequate power to address the hypothesis being tested?

-Were correct statistical analysis used to support conclusions?

-Are there concerns about ethical or regulatory requirements being met?

Reviewer #1: No further analyses likely to be helpful. Authors should list the actual numbers of the IRB approvals given in 2010 and 2013 such that the data can be linked to the specific approval documents.

Reviewer #2: Methods used mirrors similar studies, such as Chen et al 2007 

While they only use one Allele (msp1) compared to the 3 used in Chen et al, the high res seq used compensates for this

Reviewer #3: (No Response)

**Results**

-Does the analysis presented match the analysis plan?

-Are the results clearly and completely presented?

-Are the figures (Tables, Images) of sufficient quality for clarity?

Reviewer #1: Analysis very similar to same authors 2015 JID paper from Cambodian parasites and appears to be sufficient.

Reviewer #2: Data presented sufficient, however would benefit form stratification if soldiers suffered acute vivax in Papua

Reviewer #3: (No Response)

**Conclusions**

-Are the conclusions supported by the data presented?

-Are the limitations of analysis clearly described?

-Do the authors discuss how these data can be helpful to advance our understanding of the topic under study?

-Is public health relevance addressed?

Reviewer #1: Authors' conclusions are limited by the amount of data. They largely confirm their 2015 findings except the Indonesian soldiers with only one year's exposure had fewer relapses and less diverse parasites than those they observed in Cambodia.

Reviewer #2: conclusions are generally supported by the data, however it is important to have some clarity on the deployment infection rates

Reviewer #3: (No Response)

**Editorial and Data Presentation Modifications?**

Reviewer #1: (No Response)

Reviewer #2: (No Response)

Reviewer #3: (No Response)

**Summary and General Comments**

Reviewer #1: The manuscript repeats similar data reported by most of the same authors in JID 2015 (reference 13) from an endemic population in Cambodia. The current study's unique aspect is studying Indonesian soldiers who were exposed for a limited time period and then removed from the endemic area. As such, the general conclusion that there were fewer hypnozoites with less genetic variability fits the expected biology and is an important albeit unexciting observation.

I have a few questions that if the authors could answer and possibly add to the discussion would strengthen the manuscript.

There were two Indonesian battalions (Lumajang and Sragen) with very different (3:1) attack rates. Were there sufficient numbers to analyze the two groups separately as it would appear the former group (presumably 2010) was more highly exposed to infection during their year in Papua? If so, this would suggest that MOI in the Sragen Battalion should be lower if it is truly a surrogate for hypnozoite number.

More than a decade has elapsed since these samples were obtained. Although genotyping is slow business, successive Indonesian battalions have since been enrolled in vivax relapse studies (e.g. tafenoquine). Are there any epidemiological data to support the authors' conclusion that infection / exposure is directly reflected in the number of hypnozoites / relapses observed in such units when removed from the endemic area?

The authors list a number of possible factors that might influence vivax relapses. One of the basic ones is hepatocyte senescence since the hypnozoites cannot exist outside of liver cell. As the average life span of hepatocytes is about 1 year, it would seem likely that older hypnozoites are presented with an 'activate or die' decision during the observation period that the authors were following the Indonesian soldiers. Was there any evidence of a 'burst' of activation at the end of the study period? Alternative suggestions as to explain why or how vivax infections eventually 'burn out'?

The authors generally conclude that the hypnozoites were activating independently while the alternative hypothesis is that successive malaria infections trigger the next relapse. Are these mutually compatible ideas on hypnozoite biology or do the authors just think that their population was different because they were no longer being exposed to current malaria infections?

Reviewer #2: This is an important set of data that further contributes to our understanding of the relapse biology of P. vivax infections. The power of this study is due to its use of a relatively homogenous ‘non-immune’ cohort (presumably mostly 18-30yrs male, healthy (this info was not given)) with a limited exposure to P. vivax infection (~1year in Papua) before returning home to a non-endemic setting (Java). The high degree of homologous relapse and the postulated arguments for why this occurs make sense. 

I would be grateful if the authors considered replying to the following questions and concerns

Major question: Did the soldiers receive any prophylactic treatment in Papua (i.e. Doxy?) and if not; did they have any acute infections in Papua (especially given the extended period of deployment one would expect at least one infection).. and if so, what was used to treat the acute episode. This type of information is important as this type of data may be necessary to stratify the time to relapse data (as Chen 2007 has previously reported)

Minor comments: 

1. It would be good to describe the cohort (i.e sex, age)

2. Was any consideration given to using multiple alleles (or even a Microsat marker)rather than just one (msp) to give more confidence of clonality?

Reviewer #3: Major comments

The authors applied targeted amplicon deep sequencing to study the genetics of relapsing P. vivax parasites among Indonesian soldiers returning from malaria areas after a 12-month deployment. The study provides new insights into the biology of hypnozoite activation causing relapses. The most interesting finding is the decreasing multiplicity of infection over time, consistent with hypnozoite depletion resulting from activation and other mechanisms. Although expected, this is the first report from patients. The manuscript is well written. 

One weakness of the study is the use of single amplicon deep sequencing. The authors listed this as one of the limitations of the study and commented that this may limit the power of detecting minor variants and underestimate polyclonality. They then argued that given the genetic diversity revealed at pvmsp1, this difference would be incremental and not change overall findings. However, the level of genetic diversity revealed at pvmsp1 was relatively low with only 28 haplotypes detected from a set of 127 samples tested. This is in contrast to high genetic diversity levels reported from sample sets obtained from people living in high transmission settings where more haplotypes than sample numbers tested are commonly seen, especially in P. vivax infections. I assume the authors’ notion of high genetic diversity in this set of relapsing samples was based on the HE value of 0.77. However, this HE value does not seem to be correct based on 28 unique haplotypes obtained from 127 samples tested. Could the authors please describe how the HE value of 0.77 was derived? 

As the study participants were randomized into primaquine and no primaquine arms, it would be interesting to compare HE and MOI values between the two arms at 1st and 2nd relapses. This may reveal any difference in hyponozoite load between the two groups. 

Minor Comments

1. Abstract: “targeting amplicon sequencing” should be “targeted amplicon sequencing”.

2. Introduction- 2nd paragraph: “burden of P. vivax in Southeast Asia [3,4]” should be changed to “Southeast Asia and South Pacific” as one of the references reports findings from PNG. 

3. Results-Clonal patterns of relapsing P. vivax: “just 24% (8/33) of individuals had only one haplotype detected across both of their relapses, while 76% (21/33) had multiple haplotypes present across both relapses.” It would be clearer to change this to “just 24% (8/33) of individuals had the same haplotype detected across both of their relapses, while 76% (21/33) had different haplotypes between two relapses.”

4. Discussion: the first interesting observation of the high proportion of monoclonal parasites in relapse samples indicates individual hypnozoite clones activate independently of other clones present in the sample individual. This finding is consistent with findings of reference 8. It will strengthen the discussions by comparing the study findings and similarities between the two populations.

PLOS authors have the option to publish the peer review history of their article (what does this mean?). If published, this will include your full peer review and any attached files.

Reviewer #1: No

Reviewer #2: No

Reviewer #3: No

Figure Files:

Data Requirements:

Reproducibility:

References

---

## [Editor Report · Decision Letter 1]

9 Jul 2022

Dear Mr. Carey-Ewend,

We are pleased to inform you that your manuscript 'Hypnozoite depletion in successive Plasmodium vivax relapses' has been provisionally accepted for publication in PLOS Neglected Tropical Diseases.

Best regards,

Kevin Shyong-Wei Tan

Associate Editor

Mary Lopez-Perez

Deputy Editor

The authors have adequately addressed all the comments and concerns by the reviewers.

---

## [Editor Report · Acceptance letter]

19 Jul 2022

Dear Mr. Carey-Ewend,

We are delighted to inform you that your manuscript, "Hypnozoite depletion in successive *Plasmodium vivax* relapses," has been formally accepted for publication in PLOS Neglected Tropical Diseases.

Best regards,

Shaden Kamhawi

co-Editor-in-Chief

Paul Brindley

co-Editor-in-Chief
